# Mn(I)-catalyzed sigmatropic rearrangement of β, γ-unsaturated alcohols

Can Yang[1], Xiaoyu Zhou[2], Lixing Shen[1], Zhuofeng Ke [2]✉, Huanfeng Jiang [1] & Wei Zeng [1]✉

Sigmatropic rearrangement provides a versatile strategy to site-selectively reorganize carbon-skeleton with high atom- and step-economy. Herein, we disclose a Mn(I)-catalyzed sigmatropic rearrangement of β, γ-unsaturated alcohols via C-C σ bond activation. A variety of α-aryl-allylic alcohols and α-aryl-propargyl alcohols could undergo in-situ 1,2- or 1,3- sigmatropic rearrangements to allow for converting to complex structural arylethyl- and arylvinyl-carbonyl compounds under a simple catalytic system. More importantly, this catalysis model can be further applied to assemble macrocyclic ketones through bimolecular [2n + 4] coupling-cyclization and monomolecular [n + 1] ring-extension. The presented skeleton rearrangement would be a useful tool complementary to the traditional molecular rearrangement.

Molecular rearrangements exist ubiquitously in modern synthetic chemistry, providing a powerful strategy to reorganize complex structures in an atom- and step-economic process through one-step chemical transformation[1,2]. Almost a century ago, the Beckmann rearrangement[3,4], semi-pinacol rearrangement[5–7], Smiles rearrangement[8–12], Wolff rearrangement[13–17], and others[18] have been successively developed to allow for diversified group migration via parallel moving pattern (Fig. 1a). Meanwhile, the Cope and Claisen rearrangement[19–21] and Witting rearrangement[22,23] represent another type of cyclic transition state-based [3,3′]- and [2,3′]-sigmatropic shift, one of the most important features of these rearrangements involves synergetic C–X σ bond (X = C, O, N, etc.) formation and double-bond migration at pericyclic reaction-sites (Fig. 1b). To date, these classical group migratory and sigmatropic rearrangements have shown extremely potential in strategic synthesis of natural products, pharmaceuticals, and material molecules[24–26]. However, it is surprising that 1,2-sigmatropic rearrangement (1,2-STR) or 1,3-sigmatropic rearrangement (1,3-STR), which only involves two different reaction-sites, has remained unexplored. This protocol will probably establish an efficient platform to enable in-situ U-turn-like rearrangement, which refers to a U-shaped turn made by a molecular skeleton so as to head in the opposite direction from its original course.

Although C–C σ bond activation belongs to a significantly challenging transformation due to the high C–C bond dissociation energy and kinetical inertia[27,28], C–C σ bond functionalization has proven to be a straightforward approach to modifying complex carbon skeletons[29–40]. Unfortunately, the current unstrained C–C σ bond activation strategies generally lead to the loss of another carbon-containing component derived from C–C bond cleavage[28,33]. Thus, the development of unstrained C–C σ bond functionalization via rearrangement with high atomic and step-economy remains unmet challenges: (i) the efficient carbon–carbon activation-based group-switch generally requires 2–5 equivalents of coupling reagents[36,37,39,40], but the stoichiometric ratio of the cleaved moieties and the parent carbon skeletons is only up to a maximum of 1; (ii) the cleaved and lost carbon-containing moieties generally possess high reactivity and suffer from rapid decompositions either by processes of nucleophilic attack or by oxidation[35–37], leading to significant difficulty in their re-utilization. Nevertheless, given that ligand-directed C–C bond activation generally involves a cyclometalation process, which possibly once again traps the unsaturated carbonyl, vinyl or alkynyl species derived from C–C bond cleavage through the delicate balance between substrate-based reactivity and metal-based catalytical activity. Here, we show a Mn(I)-catalyzed carbon-skeleton reorganization of α-aryl-β, γ-unsaturated alcohols via C–C σ bond activation-based sigmatropic rearrangement (Fig. 1c).

---

[1]Key Laboratory of Functional Molecular Engineering of Guangdong Province, School of Chemistry and Chemical Engineering, South China University of Technology, 510641 Guangzhou, China. [2]School of Materials Science and Engineering, PFCM Lab, Sun Yat-sen University, 510275 Guangzhou, China.
✉e-mail: kezhf3@mail.sysu.edu.cn; zengwei@scut.edu.cn

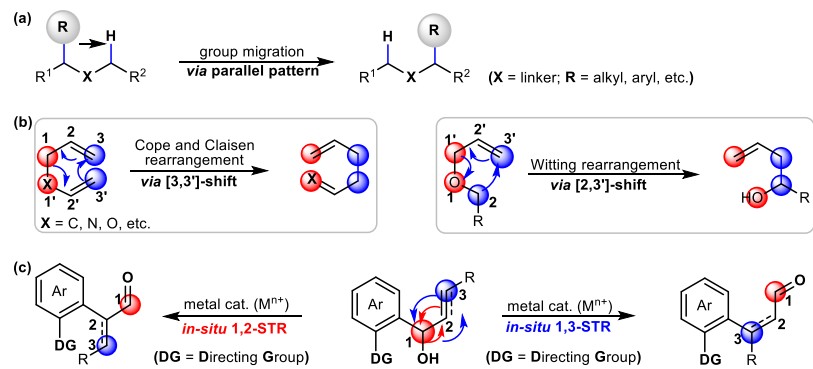

**Fig. 1 | Strategies to access molecular skeleton rearrangement. a** The traditional migratory patterns in Smiles, semi-pinacol, Beckmann, and Wolff rearrangements. **b** The sigmatropic shift in the Cope and Claisen rearrangement and the Witting rearrangement. **c** This work: Mn(I)-catalyzed 1,2-STR or 1,3-STR of α-aryl-β, γ-unsaturated alcohols.

## Table 1 | Reaction development[a]

| Entry | Catalysts | Temperature (°C) | Solvents | Yield (2a/3a) (%)[b] |
|---|---|---|---|---|
| 1 | $MnCl_2$ | 25 | PhMe | 0/0 |
| 2 | $Mn(OTf)_2$ | 25 | PhMe | 0/0 |
| 3 | $Mn(OAc)_3 \cdot 2H_2O$ | 25 | PhMe | 0/0 |
| 4 | $Mn_2(CO)_{10}$ | 25 | PhMe | 0/23 |
| 5 | $Mn(CO)_5Br$ | 25 | PhMe | 3/27 |
| 6 | $Mn(CO)_5Br$ | 45 | PhMe | 4/56 |
| 7 | $Mn(CO)_5Br$ | 75 | PhMe | 7/61 |
| 8 | $Mn(CO)_5Br$ | 85 | PhMe | 5/54 |
| 9 | $Mn(CO)_5Br$ | 75 | $PhCF_3$ | 0/43 |
| 10 | $Mn(CO)_5Br$ | 75 | THF | 0/17 |
| 11 | $Mn(CO)_5Br$ | 75 | DMF | 0/38 |
| 12 | $Mn(CO)_5Br$ | 75 | $CH_3CN$ | 0/0 |
| 13 | $Mn(CO)_5Br$ | 75 | DCE | 0/80 |
| 14 | $Cp*Co(CO)I_2$ | 75 | DCE | 0/5 |
| 15 | $Cp*Rh(CH_3CN)_3(SbF_6)_2$ | 75 | DCE | 47/25 |

[a]All the reactions were performed using **1a** (0.20 mmol) and catalysts (0.01 mmol, 5 mol %) in solvent (2.0 mL) by heat for 24 h under Ar atmosphere in a sealed tube, followed by flash chromatography on $SiO_2$.
[b]The yields were determined by isolation.

## Results

### Investigation of reaction conditions

Earth-abundant first-row metal catalysts, which possess low cost, excellent sustainability, and environmentally benign properties, have attracted increasing attention in modern synthetic chemistry[41–46]. Nevertheless, the exploration of non-noble-transition metal-catalyzed carbon-skeleton rearrangements is very scarce[47,48]. The choice of Mn-catalysts was motivated by the notion that the valence electron configuration ($3d^5 4s^2$) of elemental manganese endows different oxidation states of manganese species with distinctive coordination to arenes, alkenes, and alkynes[49,50]. We therefore utilized 1-(1-(pyridin-2-yl)−1H-indol-2-yl)prop-2-en-1-ol (**1a**) as a model substrate to evaluate the feasibility of Mn-catalyzed [1,3]-sigmatropic rearrangement via carbon−carbon sigma bond activation (Table 1). First, various manganese salts such as $Mn(OTf)_2$, $Mn_2(CO)_{10}$, and $Mn(CO)_5Br$, etc. were

investigated in toluene to optimize the catalyst system (entries 1-5), we were quite pleased to find that the treatment of substrate **1a** with $Mn_2(CO)_{10}$ and $Mn(CO)_5Br$ at room temperature (25 °C) did furnish 23% and 27% yield of 3-(1H-indol-2-yl)propanal **3a** (entries 4 and 5), respectively, in which the allylic-alcohol moiety of **1a** underwent a 1,3-STR through an in-situ carbon-skeleton-rearrangement; Meanwhile, 2-unsubstituted indole **2a**, which derived from the carbon−carbon bond cleavage and protonation, could also be obtained in 3% yield (entry 5). To our delight, after various reaction temperatures were scouted by employing $Mn(CO)_5Br$ as catalysts (entries 5–8), it was found that running this transformation at 75 °C brought us increased conversion, affording 61% yield of 1,3-STR product **3a** and 7% yield of **2a** (entries 5–8 vs. 7). More satisfactorily, screening solvent systems further confirmed that 1,2-dichloroethane (DCE) showed a positive effect on this reaction, significantly increasing the reaction yield to

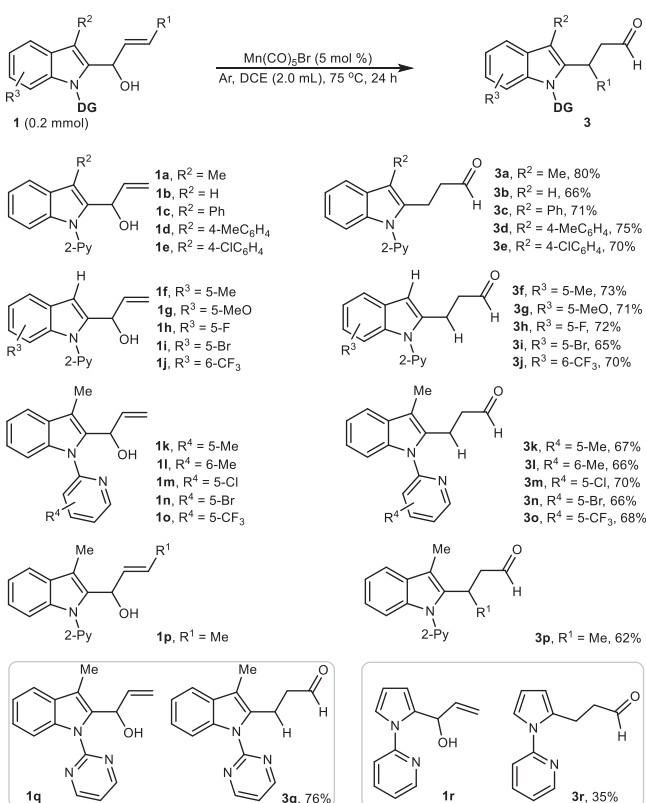

**Fig. 2 | Secondary allylic-alcohol scope.** All the reactions were performed in a sealed tube, followed by flash chromatography on SiO₂. The yields were determined by isolation.

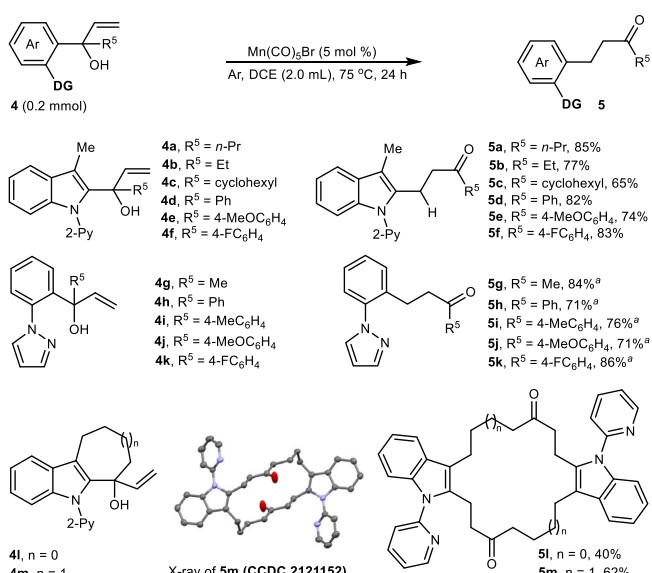

**Fig. 3 | Tertiary allylic-alcohol scope.** All the reactions were performed in a sealed tube, followed by flash chromatography on SiO₂. The yields were determined by isolation. [a]The reaction temperature is 100 °C.

80% with excellent chemoselectivity without the formation of **2a** (entries 7–12 vs. 13). Notably, Cp*Co(CO)I₂ and Cp*Rh(CH₃CN)₃(SbF₆)₂ catalysts surveyed under the optimized conditions delivered **3a** with very poor reaction conversions and chemoselectivity (entries 14 and 15).

## Substrate scope

With established reaction conditions, the α-(2-indolyl)-substituted secondary allylic alcohols were evaluated by using the optimized conditions. As summarized in Fig. 2, comparison with electroneutral indolyl-substituted allylic-alcohol **1b**, Mn(I)-catalyzed [1,3]-sigmatropic rearrangement of α-[2-(3′-alkylindolyl)] and α-[2-(3′-arylindolyl)]-substituted allylic alcohols including **1a** and **1c-1e** could efficiently produce 2-carbonylethylindoles **3a-3e** in good to excellent yields (66–80%), regardless of the steric hindrance from C3-substituents of indolyl ring (entry 1). Meanwhile, electron-donating 5-Me, 5-MeO, 5-halo-, and even strong electron-withdrawing 6-CF₃-indole-based allylic alcohols **1f-1j** in which indolyl C3-position kept unsubstituted, could also successively undergo carbon–carbon activation and 1,3-STR to provide **3f-3j** in 65–73% yields (entry 2). Moreover, N-electron-rich and N-electron-deficient pyridyl-substituted α-(2-indolyl)alcohols (**1k-1o**) were also well-tolerable in this transformation to assemble β-(2-indolyl)propanals **3k-3o** with good conversions (66-70%) (entry 3). Again, switching terminal allylic alcohols to the internal allylic-alcohol **1p** could still furnish the corresponding rearrangement product **3p** in 62% (entry 4). Of noted, α-(N-(2-pyrimidyl)indolyl) allylic-alcohol **1q** and α-(N-(2-pyridyl)pyrrolyl) allylic-alcohol **1r** also participated in this carbon-skeleton rearrangement, providing **3q** (76%) and **3r** (35%), respectively (entry 5).

In comparison with the photocatalyzed 1,3-alkyl shift of tertiary allylic alcohols reported by Knwoles[51], our catalysis system could be applied to α-aryl-tertiary allylic alcohols **4**, which underwent chemoselective 1,3-aryl transposition (Fig. 3). Among them, α-(2-indolyl)-α-alkyl-allylic alcohols (**4a-4c**) and α-(2-indolyl)-α-aryl-allylic alcohols (**4d-4f**) worked well to deliver 2-indolyl-tethered ketones **5a-5f** in 65–85% yields. Moreover, switching α-(2-indolyl)- allylic alcohols to α-phenyl-allylic alcohols (**4g-4k**) in which pyrazole was utilized as a directing group, could also produce the corresponding 2-phenyl-tethered ketones **5g-5k** in excellent reaction conversions (71-86%). Importantly, the potential breadth of the utility of this methodology is further illustrated with different six- and seven-membered α-vinyl cycloalkylalcohols **4l** and **4m**, which are effective substrates for assembling sixteen- and eighteen-membered macrocyclic molecules **5l** (40%) and **5 m** (62%, CCDC 2121152) via a carbon–carbon σ bond cleavage-based bimolecular [2n + 4] coupling-cyclization (see Supplementary Fig. 7 for the possible reaction mechanism of the formation of **5l** and **5m**).

Encouraged by Rh(I)-catalyzed 1,3-alkynyl shift of alkynyl alkenyl carbinols in which Csp³-Csp bond cleavage occurred[52], we further evaluated the rearrangement reactivity of different α-(2-indolyl) propargyl alcohols (Fig. 4). Interestingly, Mn(I)-catalyzed 1,2-STR of internal propargyl alcohols (**6a-6f**) could occur to produce α-indolyl-α-vinyl aldehydes **7a-7f** in 55-84% yields, and the carbon-skeleton rearrangement model belongs to 1,2-STR instead of 1,3-STR. Among them, the exact structure of **7f** (CCDC 2121154) was determined by its single-crystal X-ray diffraction. However, if terminal propargyl alcohol **6g** was subjected to the same reaction system, the desired 1,3-STR product 3-(2-indolyl)-propargyl aldehyde **7g** (38%) was produced possibly due to the absence of steric hindrance from the terminal substituent of propargyl alcohols. In comparison, switching terminal propargyl alcohol **6g** to the substrate **6h** with a larger size of methyl group at the alkyne terminus produced both 1,2-STR product **7h** (35%) and 1,3-STR product **7h′** (62%, E/Z = 1:3.3)[53]. Again, we explored the reactivity of α-(2-indolyl)-α-alkyl-propargyl alcohols (**6i** and **6j**) and α-(2-indolyl)-α-aryl-propargyl alcohol **6k**, and found these tertiary alcohols still smoothly underwent regioselective 1,2-STR to provide the corresponding α-(2-indolyl)-β-alkyl-α, β-unsaturated ketones **7i** (66%), **7j** (48%) and α-(2-indolyl)-β-phenyl-α,β-unsaturated ketone **7k** (77%), respectively. Similarly, 1-(2-indolyl)−3-phenylpropargyl alcohols with different substituents in the indolyl C-5 position were also amenable to this transformation, producing the desired α-(2-indolyl)-allylic aldehydes **7l**–**7q** in 38–60% yields, in which the substituents

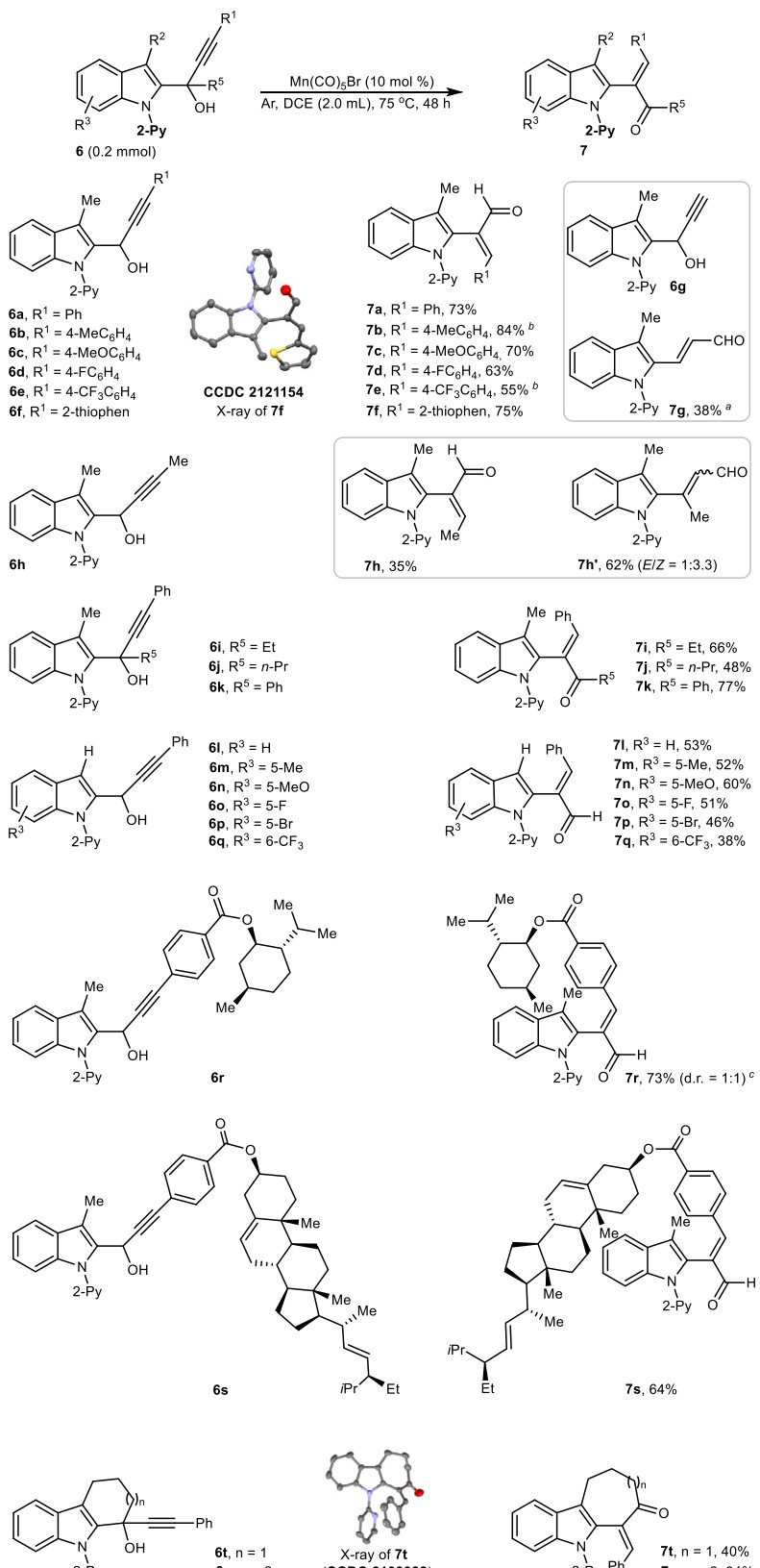

**Fig. 4 | Propargylalcohol scope.** All were performed in a sealed tube, followed by flash chromatography on SiO$_2$. The yields were determined by isolation. [a]The reaction time is 24 h. [b]The reaction time is 72 h. [c]Diastereomeric ratio (*d.r.*) was measured by $^1$H NMR spectrum.

including methyl- (**6m**), methoxyl- (**6n**), fluoro- (**6o**), bromo- (**6p**), trifluoromethyl- (**6q**) showed an apparently electronic effect on the carbon-skeleton rearrangement. More complex alkoxycarbonylphenyl-tethered propargyl alcohols, which were derived from the natural menthol (**6r**) and stigmasterol (**6s**), could still undergo intramolecular 1,2-STR to deliver the corresponding indole-containing menthol and stigmasterol derivatives **7r** (73%) and **7s** (64%), respectively. Importantly, it should be pointed out that the Mn(I)-

catalyzed 1,2-STR of α-alkynyl cycloalkylalcohols **6t** and **6u** led to a monomolecular [n + 1] ring-extension instead of bimolecular [2n + 4] coupling-cyclization, furnishing seven-membered 2-benzylidenecycloheptan-1-one **7t** (40%, CCDC 2133328) and eight-membered 2-benzylidenecyclooctan-1-one **7u** (34%), respectively.

## Application

To showcase our system's synthetic utility (Fig. 5), the pyrimidyl moiety from aldehydes **3q** could be readily removed to access free (N-H) β-(2-indolyl)ethyl-1,3-dioxolane **8** (75%). Moreover, the indole-derived aldehydes **3a** and **7a** could be further utilized as versatile platform-molecules for diversity-oriented assembly of 4-(2-indolyl)alkyne **9** (71%), 4-(2-indolyl)−5-phenylpenta-2,4-dienenitrile **10** (37%) and 2-(2-indolyl)−1,4-diphenylbuta-1,3-diene **11** (67%) by coupling with α-diazophosphonate, acetonitrile and diethyl benzylphosphonate, respectively. Of course, α, β-unsaturated aldehyde **7a** could be regioselectively reduced by NaBH₄ to give 2-(2-indolyl)−3-phenyl-allylic alcohol **13** (82%). Interestingly, the reaction between aldehyde **7a** and Me₃SiOK still provided unexpected 2-phenethyl-indole **12** (44%) through reductive deformylation. Finally, 4-(2-indolyl)pyrazole **14** (42%) was obtained after subjecting **7a** to the coupling-cyclization with phenylhydrazine.

## Mechanistic investigations

To gain insights into the mechanism of the Mn(I)-catalyzed sigmatropic rearrangement, we performed the carbon-skeleton rearrangement of **1a** with D₂O under the standard reaction system, and obtained *d*-**2a** (77%) in which 45% D was incorporated into the α-position of aldehyde (Fig. 6a), indicating that either keto-enol isomerization or protonation of carbon-metal bonds was involved in this transformation (the deuteration of the product **3a** was also performed in the presence of D₂O (5.0 equiv.) under the standard conditions, and 7.5% D was observed at the α-position of aldehyde **3a**, please see SI for more details). Evaluating the reactivity of α-(2-(*N*-phenylindolyl)allyilic alcohol **1u** as a substrate only gave an intramolecular cyclization product **3u′** (61%) instead of 1,3-STR product **3u** (Fig. 6b), demonstrating that pyridine played a key chelation-assisted role to enable this carbon-skeleton rearrangement.

Comparison with the reactivity of allylic-alcohol **1a** and 2-unsubstituted *N*-pyridylindole **7** under the same reaction conditions (Fig. 6c vs. 6d), it was found that Mn(I)-catalyzed Csp²-H bond cross-coupling of **7** with acrylaldehyde at 45 °C could only give **3a** with poor conversion (75% vs. 15%). These observations pointed to an understanding mechanism that 2-unsubstituted indole **7** was not a possible reaction intermediate derived from C−C σ bond cleavage of **1a**;

Meanwhile, the subsequent Mn(I)-catalyzed C-H activation and cross-coupling with acrylaldehyde were not possibly involved in this rearrangement. The intermolecular competing reactions between **4a** and **1g** produced 1,3-STR products **5a** (64%), **5n** (31%), **3c** (43%) and **3a** (24%), indicating that cyclomanganated species and the in-situ generated acrylaldehydes probably underwent different site-selective Mn-carbonyl and Mn-alkene complexation via a step-wise process (Fig. 6e).

Based on these control experiments, the possible reaction mechanism is proposed in Fig. 7. The interaction between allylic-alcohol **1a** with Mn(CO)₅Br produced cyclomanganated species **A**, followed by chelation-assisted β-aryl elimination to give six-coordination Mn-carbonyl complexes **B**, in which the "carbonyl oxygen" of the in-situ generated acrylaldehyde coordinated to Mn(I) cations. Subsequently, Mn-aldehyde intermediate **B** underwent an intramolecular ligand exchange to give Mn-alkene complexes **C** in Path a. The subsequent migratory insertion and protonation of carbon-Mn bonds of complexes **C** afforded the 1,3-STR product **3a**. Of course, an alternative intramolecular Michael-addition initiated by Mn-aldehyde intermediate **B** via Path b could possibly proceed to furnish enol oxygen-coordinated Mn(I)-complexes **E**, which then underwent a cascade protonation and keto-enol tautomerization to afford **3a**. Instead, indole **7** and Mn-complexes **F** as possible intermediates, were not involved in this [1,3]-sigmatropic rearrangement, supported by our experimental exclusion of the C-H activation (Fig. 6d).

Density functional theory (DFT) studies were performed to explore the Mn(I)-catalyzed carbon-skeleton rearrangement (the potential free energy profiles of the major reaction pathways in Fig. 8 and the full version in Supplementary Fig. 8). The transformation begins with the cyclomanganated complex **A**. Both the *cisoid* and the *transoid* isomers of the formed enal are considered for the β-aryl elimination. The Gibbs free energy of ***transoid*-A** is 2.5 kcal/mol higher than that of ***cisoid*-A** due to the steric effect between the vinyl and the indole fragments. The release of one CO molecule from the cyclomanganated complex **A** gives intermediate **A1** (***cisoid*-A1**: 9.3 kcal/mol **vs. *transoid*-A1**: 10.6 kcal/mol), providing a vacant site for the β-aryl elimination. The chelation-assisted β-aryl elimination from ***cisoid*-A1** or ***transoid*-A1** then goes through transition states ***cisoid*-TS1** (16.2 kcal/mol) or ***transoid*-TS1** (18.5 kcal/mol), respectively, to furnish the six-coordination Mn-carbonyl intermediates **B** (***cisoid*-B**: 1.4 kcal/mol **vs. *transoid*-B**: −1.2 kcal/mol). The Gibbs free energy of activation of ***cisoid*-TS1** is slightly lower than that of ***transoid*-TS1** by 2.3 kcal/mol due to the similar steric effect. After C-C sigma bond cleavage, as expected the *transoidal* enal ***transoid*-B** (−1.2 kcal/mol) is more stable than the *cisoidal* isomer ***cisoid*-B** (1.4 kcal/mol). The Mn-carbonyl intermediates **B** could isomerize to the Mn-alkene intermediates **C**

**Fig. 5 | Synthetic applications. a** Removal of pyrimidyl group. **b** Alkynylayion of aldehydes. **c** Cyanation of aldehydes. **d** Arylvinylation of aldehydes. **e** Reduction of carbon−carbon double bonds and carbonyl group from α, β-unsaturated aldehydes. **f** Reduction of aldehydes. **g** Coupling-cyclization of α, β-unsaturated aldehydes with hydrazines.

**Fig. 6 | Preliminary mechanism studies. a** H/D exchange experiment. **b** Intramolecular cyclization of α-[2-(*N*-phenyl) indolyl]allylic-alcohol. **c** Csp²-Csp³ σ bond activation-directed carbonylethylation of indoles. **d** Csp²-H bond activation-directed carbonylethylation of indoles. **e** Mn(I)-catalyzed cross-coupling reaction via 1,3-STR.

**Fig. 7 | Proposed reaction mechanism.** Path a Mn(I)-catalyzed 1,3-STR via intramolecular ligand exchange and migratory insertion. Path b Mn(I)-catalyzed 1,3-STR via intramolecular Michael-addition.

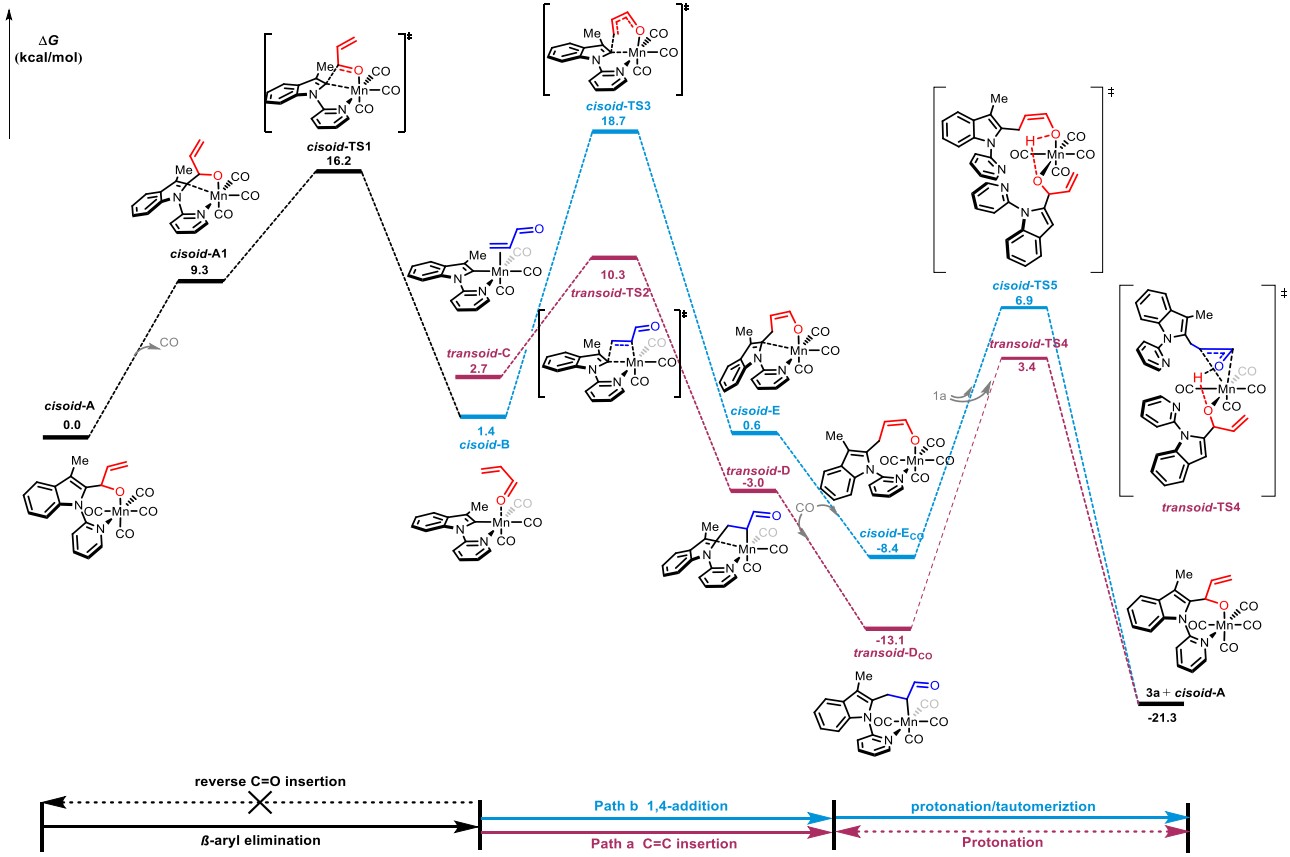

**Fig. 8 | Computed potential energy surfaces for the major reaction pathways of the Mn(I)-catalyzed carbon-skeleton rearrangement via C−C bond activation at the SMD(DCE)/M06L/Def2tzvp//M06L/def2svp level of theory.** Purple line the DFT-computed energy surfaces of Path a. Blue line the DFT-computed energy surfaces of Path b.

(*cisoid*-**C**: 4.8 kcal/mol **vs.** *transoid*-**C**: 2.7 kcal/mol). Subsequently, the Mn-alkene intermediate **C** initiates the migratory C = C insertion to the afford cyclo Mn-alkyl intermediate **D** through the four-membered ring transition state **TS2** (Path a). Compared with *cisoid*-**TS2** (11.4 kcal/mol), *transoidal C* isomer results in a more stable migration insertion transition state ***transoid*-TS2** (10.3 kcal/mol). Nevertheless, the formed Mn-alkyl intermediate **D** will recombine the CO ligand and release product **3a** after proton transfer and keto-enol tautomerism. The OH of **1a** coordinates with the metal center of intermediate **D$_{CO}$** to promote the intramolecular proton transfer from **1a** to acrylaldehyde via transition states ***cisoid*-TS4** (4.2 kcal/mol) or ***transoid*-TS4** (3.4 kcal/mol), which then releases **3a** after keto-enol tautomerism. On the other hand, the Mn-carbonyl **B** promoted 1,4-Michael-addition (Path b) is suggested to be less favored with a calculated Gibbs free energy of 18.7 kcal/mol for transition state ***cisoid*-TS3** due to the ring constraint. Although 1,4-addition may produce the enolate oxygen-coordinated Mn(I)-intermediates *cisoid*-**E** (0.6 kcal/mol). Similarly, the introduction of **1a** to *cisoid*-**E$_{CO}$** initiates the proton transfer via transition state ***cisoid*-TS5** (6.9 kcal/mol), leading to **3a** after keto-enol tautomerism.

To further understand the driving force of the Mn(I)-catalyzed sigmatropic rearrangement, the localized Kohn-Sham orbitals and distortion/interaction analysis of the migratory insertion (C=C insertion) and the reverse β-aryl elimination (C=O insertion) were carried out. As shown in Fig. 9, the reverse β-aryl elimination reaction (C=O insertion) is driven by 1) the interaction between the C-Mn p-d orbital and the C=O π* antibonding orbital, and 2) the interaction between the C=O π orbital and the Mn dz$^2$ antibonding orbital. In comparison, the C=C insertion is contributed by the donation of the C-M p-d orbital to the C=C π* orbital and the donation of the C=C π orbital to the antibonding dz$^2$ orbital. In both cases, the energy gaps

between the C-M p-d orbital and the π* orbital (3.10 eV for the C=O insertion, 3.05 eV for the C=C insertion) are smaller than those between the π orbital and the antibonding dz$^2$ orbital (4.46 eV for the C=O insertion, 4.54 eV for the C=C insertion), suggesting the p-d to π* interaction attributes mainly for the insertion. Notably, the p-d to π* interaction is more significant in the C=C insertion (ΔE$_{gap}$ = 3.05 eV) than in the C=O insertion (ΔE$_{gap}$ = 3.10 eV), resulting in a better interaction energy (ΔE$_{int}$ = −14.3 kcal/mol) in the C=C insertion transition state ***transoid*-TS2**, which is probably the driving force of the preference of C=C insertion over C=O insertion for the rearrangement process. Furthermore, the orbital composition analysis in the LUMO of the substrates indicates that the terminal C$^4$ p orbital (32.9%) in the C=C insertion step is greater than the carbonyl C$^2$ p orbital (27.5%) in the C=O insertion step, resulting in a large driving force for the C=C insertion, guaranteeing the subsequent 1,3-STR. The difference in the interaction energies between ***transoid*-TS2** (−14.3 kcal/mol) and ***cisoid*-TS1** (−7.8 kcal/mol) thus resulted in a lower Gibbs free energy of activation for the C=C insertion transition state ***transoid*-TS2**.

## Discussion
In conclusion, we developed a Mn(I)-catalyzed carbon-skeleton rearrangement via C−C σ bond cleavage. A variety of α-aryl-β, γ-unsaturated alcohols were applicable to this simple catalytic system. This protocol features with [1,n]-sigmatropic rearrangement (n = 2, 3), leading to 1,2-STR and 1,3-STR to reorganize carbon-skeletons with high atom- and step-economy. This sigmatropic rearrangement strategy can enrich the rearrangement chemistry and enhance the development of more challenging carbon-skeleton rearrangements in the future.

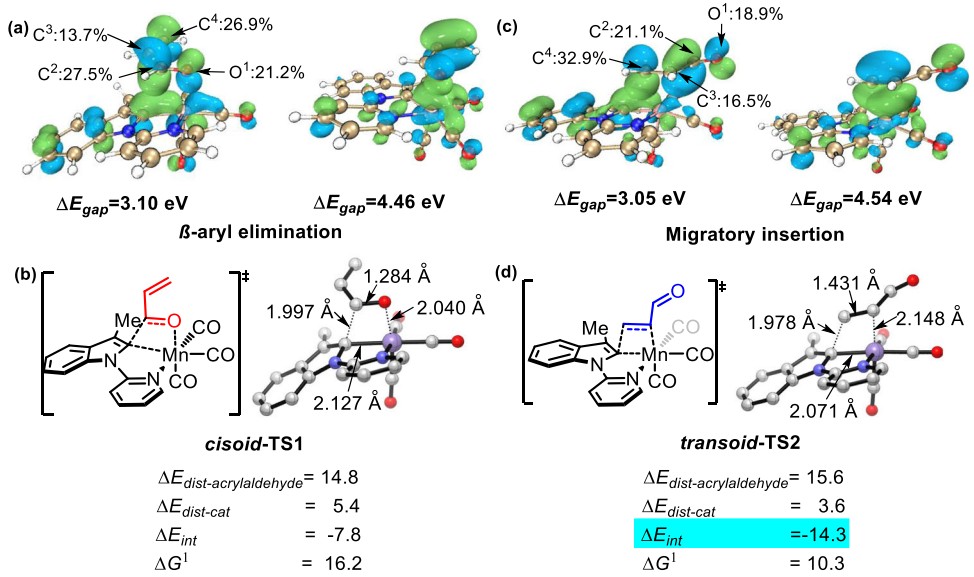

**Fig. 9 | The localized Kohn-Sham orbitals and distortion/interaction analysis.**
**a** C-Mn p-d orbital and C=O π* antibonding orbital interaction, C=O π orbital and
Mn dz² orbital interaction. **b** Deformation/interaction analysis of the transition
state for the β-aryl elimination pathway. **c** C-M p-d orbital and C=C π* orbital

interaction, C=C π orbital and Mn dz² orbital interaction. **d** Deformation/interaction
analysis of the transition state for the migratory insertion pathway. Energies are in
kcal/mol, isovalue of Kohn-Sham orbitalis is 0.05.

## Methods

### Procedure for the Mn(I)-catalyzed sigmatropic rearrangement of β, γ-unsaturated alcohols

All the reactions were run in an oven-dried 2 dram vial fitted with an oven-dried Teflon coated stir bar, and a Teflon cap under argon atmosphere. The Mn(CO)$_5$Br that was used was stored in a freezer. Before every reaction set-up, DCE was freshly distilled. To a 10 mL vial equipped with a magnetic stir bar, was added α-aryl-β, γ-unsaturated alcohols (0.2 mmol), Mn(CO)$_5$Br (5 mol %) and DCE (2.0 mL) under Ar atmosphere. The reaction mixture was then allowed to stir at 75 °C for 24 h. After the reaction mixture was cooled down, the corresponding reaction mixture was purified by flash chromatography on silica gel to afford the desired 1,2-STR and 1,3-STR products.

## Data availability

The authors declare that the data relating to the materials and methods, experimental procedures, mechanism research, NMR spectrum (the original Data-I in supplementary data file), HR-MS spectrum (the original Data-II in supplementary data file), IR spectrum (the original Data-III in supplementary data file), DFT calculations (the original Data-IV in supplementary data file), and X-ray structural analysis (the original Data-V in supplementary data file) are available within its Supplementary Information file, while the original data has been deposited on figshare [https://figshare.com/s/46dc86c784eb71d8d7c9]. The X-ray crystallographic coordinates for structures **5m**, **7f** and **7t** reported in this study have been deposited at the Cambridge Crystallographic Data Centre (CCDC) under CCDC 2121152, CCDC 2121154 and CCDC 2133328, respectively. These data can be obtained free of charge from The Cambridge Crystallographic Data Centre via www. ccdc.cam.ac.uk/data_request/cif.

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

## Acknowledgements

The authors thank the National Natural Science Foundation of China (No. 21871097, W. Z.; No. 21973113, Z.-F. K.; No. 22271100, W. Z.), the Key-Area Research and Development Program of Guangdong Province, China (No. 2020B010188001, H.-F. J.), and the Guangdong Basic and Applied Basic Research Foundation (No. 2023A1515010070, W. Z.) for the financial support.

## Author contributions

W.Z. directed the research. C.Y. and L.S. performed the experiments. Z.K. directed the DFT calculations. X.Z. performed the computational studies. W.Z. wrote the manuscript. H.J. and Z.K. contributed to discussions. (C.Y. and X.Z.) These two authors contributed equally to this work.

## Competing interests

The authors declare no competing interests.
