## [Peer Review File · Nature Communications]

Reviewers' comments:

Reviewer #1 (Remarks to the Author):

This paper describes an interesting manganese-catalyzed skeletal rearrangement reaction of indolyl-substituted allyl alcohols to 3-indolylpropanal. The reaction is applicable to a variety of allyl alcohols and yields the desired products in good yields. The fact that the reaction of 2-unsubstituted indole with acrolein is much less efficient proves the rationality of this strategy based on the skeletal rearrangement reaction. The proposed mechanism rationally explains the results obtained. The DFT calculations support this proposal well. The authors showed that the reaction 1a in the presence of heavy water yields partially deuterated products and claimed the incorporation of protonation of the C-Mo bond or isomerization of the keto-enol. However, under the conditions of this reaction, it is possible that the products underwent H/D exchange. This possibility should be considered and reported in Supporting information. If possible, the reaction of allyl alcohols to produce methacrolein is also worth considering, since the geometric preference of transition state 1 may affect the reaction rate.

In addition, the authors found that the reaction of propargylic alcohol proceeds by a formal [1,2] rearrangement of the indolyl group, except that the terminal alkyne 6g undergoes a [1,3] rearrangement to produce 7g. Substrates with alkyl groups at the alkyne terminus need to be investigated. Alkyl groups that are less bulky than phenyl groups may give a mixture of 1,2- and 1,3-rearrangements, supporting that bulkiness controls the reaction pathway. DFT calculations are also highly desirable for the reaction of propargylic alcohols to understand their interesting product selectivity.

Because this transformation is not only interesting as a new skeletal rearrangement reaction, but also useful as a synthetic method for target compounds, The reviewer recommends this manuscript for publication in Nature Communications, taking into account the aforementioned issues.

Reviewer #2 (Remarks to the Author):

The authors disclose Mn(I)-catalyzed rearrangement of indolyl-substituted allylic alcohols and propargyl alcohols via C-C bond cleavage. These reactions provide an access to indolyl ketones and indolyl conjugated ketones. There are several critical concerns that make this manuscript not suitable for publication in Nature Communications.

1. As proposed in Fig 8, these reactions are based on typical organometallic catalysis involving several elementary steps but absolutely not [1, n]-sigmatropic rearrangements.

2. The concept proposed by the authors are not correct also with respect to the term “group-inversion”. Inversion in organic chemistry could typically be used for properties such as polarity or configuration but not the position of functional groups. In fact, similar rearrangements are known but are overlooked by the authors (e.g., J. Am. Chem. Soc. 2007, 129, 46, 14158). It appears that the authors are just making new concepts with respect to analogues known rearrangement reactions.

3. Rearrangement of allylic alcohols to beta-functionalized ketones could be achieved via light-driven proton-coupled electron transfer (PCET) activation of the O–H bond (e.g., Angew Chem Int ed. 2021, 20190). It is superior to the present method as here it needs direct group and appears to be restrained to indole substrates (no other type of substrates are shown), thus strongly compromising the soundness and also the synthetic potential of this methodology.

4. The Fig 1c and accompanying text are misleading, only showing the carbon skeleton arrangement. However, the real substrates are allylic alcohols and propargyl alcohols.

5. The E/Z ratios of products shown in Fig 5 should be provided and discussed. Also, the E/Z configuration is inverted from 7a to 10, is it correct?

6. The formation of macrocyclic molecules 5g and 5h through bimolecular [2n + 4] coupling-cyclization should be explained in detail. Do these two cases consist with the mechanism proposed in Fig 8?

Reviewer #3 (Remarks to the Author):

In this work, the Mn(I)-catalyzed group-inversion rearrangement reaction was experimentally reported and its reaction mechanism was computationally investigated using DFT method. This reaction is interesting, but the presence of coordinating pyridyl group at the neighboring site of the reactive C-H bond seems necessary. Such condition limits the application scope of this reaction.

In addition, I found several questions in the computational part.

1) The M06L was used. The M06L is useful as well known for transition metal systems but the computational results about the activation energy have not been checked well because of the lack of

the experimental results. Because the GGA-type functional generally tends to underestimate the activation energy, I am afraid that the present activation energy is smaller than the truth. The authors must check the reliability of the DFT method with M06L. At least, they must use the hybrid-type functional with dispersion correction. The best way is to check if the DFT functional can be used for this reaction by comparing the DFT computational results with the MP4(SDQ)-computational ones using some reasonable (simplified) model.

2) The final step of the reaction (D → A in Fig. 8) is described as the protonation. If this is protonation, what acid participates in this step? Is it HBr or other acid? In the case of the other acid, what is anion?

The other plausible possibility is the reaction of the OH group of the substrate 1a with the Mn-alkyl bond of the intermediate D. In my understanding, the reaction between the OH group of 1a and the Mn-alkyl bond of the intermediate D is the most plausible elementary step. Such reaction does not occur with small activation energy. The authors must calculate the last step.

3) Also, the authors did not investigate how the active species is formed from MnBr(CO)₅ and 1a. In this step, the C-H sigma bond activation is involved. It is likely that the reaction does not occur with small activation energy. It is important to show how much easily the active species is formed.

4) The analysis shown in Fig. 10 is not reasonable. This is the insertion reaction of the C=C or C=O double bond into the Mn-alkyl bond. In Fig. 10, the authors explained that the charge-transfer (CT) from Mn d-pai orbital to the C=C (or C=O) pai* orbital is important for this transition state. However, this CT does not lead to the formation of the C-C bond. In the transition state of the insertion reaction, the sp² orbital of the vinyl group, which interacts with the Mn empty d-sigma orbital in the intermediate B, starts to interact with the pai* orbital of the C=C (or C=O) double bond, which participates in the pai-back donation interaction with the Mn atom in the intermediate B. We must consider the possibility that the activation energy would become larger if the pai-back donation is strong in the intermediate B. In other words, the strong pai-back donation is not always favorable for lowering the activation energy. The schematical figure of Fig. 10 is not appropriate for discussion and the discussion must be corrected.

5) Page 15 the last line to page 16 the first line: The orbital energy gaps discussed here show small difference (0.05 eV). In Figure 10, the Delta E(gap) is presented below the Kohn-Sham orbital at the transition state. In my understanding, this is the energy gap between the Mn occupied d orbital of the separated Mn moiety and the pai* orbital of the C=O or C=C bond of the separated substrate. The definition must be presented.

As described in my comment 4, the important orbital energy gap is that between the sp² orbital of the vinyl moiety and the pai* orbital of the C=O or C=C bonds. The authors must present such orbital energy gap.

The other factor for determining the regioselectivity is the orbital shape of the pai* orbital. The authors must compare the size of terminal C p orbital and the carbonyl C p orbital in the LUMO of the substrate. The difference in size leads to the difference in regioselectivity.

Minor comments:

Several words are not correctly used. The authors must ask the physical and/or theoretical chemist to check the sentences.

“activation free energy”; There are two kinds of free energy. Here, either “the Gibbs energy of activation” or “the Gibbs free energy of activation” is correct. Or, the Gibbs activation energy is useful, though this is not a correct name.

Back-bonding: What is the chemical bond toward back-side? The “back-donation” must be used.

Response to Referees

Journal: *Nat. Commun.*

Manuscript ID: NCOMMS-22-27391-T

Title: Mn(I)-Catalyzed In-Situ Group-Inversion Rearrangement

Author(s): Can Yang, Xiaoyu Zhou, Lixing Shen, Zhuofeng Ke*, Huanfeng Jiang, and Wei Zeng*

Address: ^a Key Laboratory of Functional Molecular Engineering of Guangdong Province, School of Chemistry and Chemical Engineering, South China University of Technology, Guangzhou 510641, China; ^b School of Materials Science and Engineering, PFCM Lab, Sun Yat-sen University, Guangzhou 510275, China

Dear Referees,

The above-mentioned manuscript was ever submitted to *Nat. Commun. (NC)* for peer review, we got the review results on Aug. 21st. Now we have already revised our manuscript according to the referees' suggestions, Some revisions are shown as follows, please check them.

A. Respond to the 1st reviewer's comments

Question 1: The authors showed that the reaction 1a in the presence of heavy water yields partially deuterated products and claimed the incorporation of protonation of the C-Mo bond or isomerization of the keto-enol. However, under the conditions of this reaction, it is possible that the products underwent H/D exchange. This possibility should be considered and reported in Supporting information.

Respond to this question: We already ran the deuteration of the product **3a** in the presence of D₂O (5.0 equiv.) under the same reaction conditions, and 7.5% D was observed at the α -position of aldehyde **3a**. We already added this result into SI (see Pg S-40, **Fig. S2**), please check it, thanks.

Question 2: In addition, the authors found that the reaction of propargylic alcohol proceeds by a formal [1,2] rearrangement of the indolyl group, except that the terminal alkyne **6g** undergoes a [1,3] rearrangement to produce **7g**. Substrates with alkyl groups at the alkyne

terminus need to be investigated.

Respond to this question: We already tried this reaction by employing 1-(3-methyl-1-(pyridin-2-yl)-1H-indol-2-yl)but-2-yn-1-ol as a substrate which bears a methyl group at the alkyne terminus, and the reaction result was shown as follows, both 1,2-U-Turn product **7h** (35%) and 1,3-U-Turn product **7h'** (62%, *E/Z* = 1:3.3) were obtained, please see the exact ¹H NMR spectrum and ¹³C NMR spectrum in our revised manuscript including SI.

Question 3: DFT calculations are also highly desirable for the reaction of propargylic alcohols to understand their interesting product selectivity.

Respond to this question: We already tried different aryl group-(**6a-6f**), alkyl group (**6h**)-substituted internal propargylic alcohols and terminal propargylic alcohol (**6g**). As you can see, different sizes of substituents (Ar, Me, and H) at the alkyne terminus already have been confirmed to affect the regioselectivity of this transformation due to their steric hindrance, so we think that it is not necessary to further perform the DFT calculations to explain this result, thanks.

B. Respond to the 2nd reviewer's comments

Question 1: As proposed in Fig 8, these reactions are based on typical organometallic catalysis involving several elementary steps but absolutely not [1, n]- sigmatropic rearrangements.

Respond to this question: In Pg104 of the book “Essentials of Pericyclic and Photochemical Reactions” of Prof. Biswanath Dinda, “*Sigmatropic rearrangements*” refers to a shift of one sigma-bonded atom or group from its allylic type position to the distant end of the conjugated π system followed by simultaneous shift of π electrons. In our manuscript, this transformation is catalyzed by transition metal catalysts, but these reactions such as **Fig. 3** and **Fig. 5** all feature with the shift of Csp²-Csp³ bonds and π electrons. Therefore, our reaction at least belongs to formal [1, n]- sigmatropic rearrangement, *via* a different mechanism.

Question 2: The concept proposed by the authors are not correct also with respect to the term “group-inversion”. Inversion in organic chemistry could typically be used for properties such as polarity or configuration but not the position of functional groups. In fact, similar rearrangements are known but are overlooked by the authors (e.g., *J. Am. Chem. Soc.* 2007, 129, 46, 14158). It appears that the authors are just making new concepts with respect to analogues known rearrangement reactions.

Respond to this question: Prof. T. Hayashi’s work (*J. Am. Chem. Soc.* 2007, 129, 46, 14158.) was involved in Rh-catalyzed **beta-alkynyl** elimination in which **Csp³-Csp** bond cleavage occurred, and then followed by Michael addition (see **Scheme 1**).

Scheme 1. Prof. T. Hayashi’s work (*J. Am. Chem. Soc.* 2007, 129, 46, 14158.)

Please pay attention to that our work was involved Mn(I)-catalyzed **beta-aryl elimination** in which **Csp²-Csp³** bond cleavage occurred, then followed by “U-Turn” rearrangement of α , β -unsaturated alcohols *via* migratory insertion (*not Michael addition pathway*) (see **Scheme 2** and **our DFT calculations**). Our catalytic system and substrates are fundamentally different from Prof. T. Hayashi’s work.

Scheme 2. Our work (see our manuscript)

As for the new concept “group-inversion”, considering that the 2nd reviewer could not accept it, we changed “group-inversion rearrangement” to “U-Turn” rearrangement in our whole manuscript, and hope the 2nd reviewer can accept the concept, thanks.

Moreover, we already changed the manuscript title “**Mn(I)-Catalyzed In-Situ Group-Inversion Rearrangement**” to “**Mn(I)-Catalyzed U-Turn Rearrangement of α , β -Unsaturated Alcohols**”, please check it, thanks.

Question 3: Rearrangement of allylic alcohols to beta-functionalized ketones could be achieved via light-driven proton-coupled electron transfer (PCET) activation of the O–H bond (e.g., *Angew Chem Int ed.* 2021, 20190). It is superior to the present method as here it needs direct group and appears to be restrained to indole substrates (no other type of substrates are shown), thus strongly compromising the soundness and also the synthetic potential of this methodology.

Respond to this question: Prof. R. R. Knowles’s work (*Angew. Chem., Int. Ed.* 2021, 60, 20190) was involved in Ir-catalyzed **alkyl shift** of **tertiary allylic alcohols** under photocatalysis system in which **Csp³-Csp³** bond cleavage and alkyl group shift occurred (see **Scheme 3**).

Scheme 3. R. R. Knowles's work (*Angew. Chem., Int. Ed.* 2021, 60, 20190)

As we know that Mn catalysts belong to cheaper catalysts than Ir catalysts. Moreover, our reaction involved **aryl shift**; Again, our reaction substrate scope includes **alpha-aryl-secondary allylic alcohols**, **alpha-aryl-propargyl alcohols**, in which the **aryl skeletons** could be extended to **indolyl skeletons**, **pyrrolyl skeletons** and **phenyl skeletons**, please see **3r**, **5g-5k** in our revised manuscript. More importantly, this reaction was involved in "U-Turn rearrangement" of allylic alcohols and propargyl alcohols. Therefore, our reaction (**Scheme 4**) is fundamentally different from R. R. Knowles's work (*Angew. Chem., Int. Ed.* 2021, 60, 20190).

Scheme 4. Our work (see our manuscript)

Question 4: The Fig 1c and accompanying text are misleading, only showing the carbon skeleton arrangement. However, the real substrates are allylic alcohols and propargyl alcohols.

Respond to this question: We already noted that R¹ prefers to OH-group, and redrew the corresponding reaction scheme as follows, please check **Fig. 1c**, thanks.

Fig. 1c. This work: *In-Situ* U-Turn rearrangement.

Question 5: The *E/Z* ratios of products shown in Fig 5 should be provided and discussed. Also, the *E/Z* configuration is inverted from **7a** to **10**, is it correct?

Respond to this question: We already got the single crystal structures of products **7f** and **7t** which belong to *E* configuration, so we assigned all the products **7a-7t** with *E* configuration, please check Fig. 5.

By the way, the configurations of **10** and **13** in **Fig. 6** were wrong, we already corrected it as follows, thanks.

Fig. 6. Synthetic applications. (a) Removal of pyridyl group. (b) Alkynylation of aldehydes. (c) Cyanation of aldehydes. (d) Arylvinylation of aldehydes. (e) Reduction of carbon-carbon double bonds and carbonyl group from α , β -unsaturated aldehydes. (f) Reduction of aldehydes. (g) Coupling-cyclization of α , β -unsaturated aldehydes with hydrazines.

Question 6: The formation of macrocyclic molecules **5g** and **5h** through bimolecular $[2n + 4]$ coupling-cyclization should be explained in detail. Do these two cases consist with the mechanism proposed in Fig 8?

Respond to this question: We already explained the possible reaction mechanism as follows, please see SI and check the Fig. S7 in Pg S-46.

...Meanwhile, the possible mechanism for the formation of **5g** (**5g** was already changed to **5l** in our revised manuscript) and **5h** (**5h** was already changed to **5m** in our revised manuscript) was also shown in Fig. S6, in which the interaction between allylic alcohol **4l** or **4m** with $\text{Mn}(\text{CO})_5\text{Br}$ produced cyclomanganated species **A-4**, followed by chelation-assisted β -aryl elimination to give six-coordination Mn-carbonyl complexes **B-4**. Subsequently, bimolecular migratory insertion of Mn-intermediate **B-4** and protonation of carbon-Mn bonds of complexes **B-5** afforded the macrocyclic products **5l** and **5m**.

Fig. S7. Proposed reaction mechanism for the formation of 5l and 5m

C. Respond to the 3rd reviewer's comments

Question 1: The M06L was used. The M06L is useful as well known for transition metal systems but the computational results about the activation energy have not been checked well

because of the lack of the experimental results. Because the GGA-type functional generally tends to underestimate the activation energy, I am afraid that the present activation energy is smaller than the truth. The authors must check the reliability of the DFT method with M06L. At least, they must use the hybrid-type functional with dispersion correction. The best way is to check if the DFT functional can be used for this reaction by comparing the DFT computational results with the MP4(SDQ)-computational ones using some reasonable (simplified) model.

Respond to this question: Thanks for your comments. We have recalculated the key intermediates and transition states using M06-L, M06-L(D3), M06(D3), B3LYP(D3), PBE0(D3), and MN15 functional. All the results lead to the same conclusion that the Gibbs activation energies of the β -aryl elimination step via *cisoid-TS1* or *transoid-TS1* are higher, to the same extent, than those of the 1,4-addition and the C=C insertion steps (see table below). For the β -aryl elimination step, the Gibbs activation energies with the hybrid-type functional B3LYP(D3) are close to that with M06-L, M06-L(D3), M06(D3) functional, while the activation energies with the hybrid-type functional PBE0-D3 and MN15 are only slightly higher than that with M06L functional. The DFT computational results with M06-L, M06-L(D3), M06(D3), B3LYP(D3), PBE0(D3), and MN15 functional is consistent with the experimental results. Due to the expense of the MP4(SDQ) method in our systems, we could not obtain the results presently, however, these results suggest different DFT methods led to the same conclusion and did not influence our discussion of the mechanism. This discussion has been added in SI.

Table S8. The comparison of the Gibbs free energies of the key intermediates and transition states using different functions

	M06-L	M06-L(D3)	M06(D3)	B3LYP(D3)	PBE0(D3)	MN15
cisoid-A	0.0	0.0	0.0	0.0	0.0	0.0
transoid-A	2.5	2.5	0.4	0.9	0.6	1.6
cisoid-TS1	16.2	16.6	16.3	16.1	19.0	18.2
transoid-TS1	18.5	18.8	17.2	17.7	20.2	19.4
cisoid-B	1.4	1.1	1.4	-0.9	3.3	4.5
transoid-B	-1.2	-0.4	0.0	-3.4	3.7	3.8
cisoid-C	4.8	5.2	2.1	4.3	4.2	4.6
transoid-C	2.7	3.2	0.7	2.1	3.7	2.7
cisoid-TS2	11.4	10.6	8.0	11.4	13.3	12.3
transoid-TS2	10.3	11.9	7.0	9.7	12.4	10.9
cisoid-TS3	18.7	18.8	19.2	15.4	25.5	23.0

Question 2: The final step of the reaction (D to A in Fig. 8) is described as the protonation. If this is protonation, what acid participates in this step? Is it HBr or other acid? In the case of the other acid, what is anion?

Respond to this question: In Fig. 8, for the protonation, the substrate alcohol itself could sever as a proper proton source. In addition, the trace HBr derived from the starting material alcohols **1a** *via* ligand exchange, could also be possibly involved in this process.

For the final step of the reaction D to A, we calculated the reaction that the alcohol substrate **1a** as the proton source. It is the intramolecular proton transfer between the OH group of **1a** and

the Mn-alkyl intermediate D_{CO} or the enolate oxygen-coordinated Mn(I)-intermediates *cisoid-E_{CO}* (as shown in **Fig.9**, the calculated detail shown in respond of question 3). Then release **3a** after keto-enol tautomerism. Their calculated Gibbs free energies of activation are about 15.3~16.6 kcal/mol. However, when the HBr act as the proton source, the intermediates *cisoid-A_{pre}* (-13.4 kcal/mol) or *transoid-A_{pre}* (-12.1 kcal/mol) was formed. With the assistance of the bromide, the alcohol is deprotonated via transition states *cisoid-TS6* or *transoid-TS6*, respectively (the detail of this process shown in respond of question 4). The Gibbs free energies of activation are about 19.2~21.7 kcal/mol, which is higher than that of the alcohol substrate **1a** as the proton source reaction.

Question 3: The other plausible possibility is the reaction of the OH group of the substrate **1a** with the Mn-alkyl bond of the intermediate **D**. In my understanding, the reaction between the OH group of **1a** and the Mn-alkyl bond of the intermediate **D** is the most plausible elementary step. Such reaction does not occur with small activation energy. The authors must calculate the last step.

Respond to this question: Thanks for your suggestion. We calculated the last step, which is the intramolecular proton transfer reaction between the OH group of **1a** and the Mn-alkyl intermediate D_{CO} or the enolate oxygen-coordinated Mn(I)-intermediates *cisoid-E_{CO}*. The located transition states have been added to **Fig.9**. Their calculated Gibbs free energies of activation are about 15.3~16.6 kcal/mol.

We have also added more discussion to the revised manuscript “Nevertheless, the formed Mn-alkyl intermediate **D** will recombine the CO ligand and release product **3a** after proton transfer and keto-enol tautomerism. The OH of **1a** coordinates with the metal center of intermediate D_{CO} to promote the intramolecular proton transfer from **1a** to acrylaldehyde via transition states *cisoid-TS4* (4.2 kcal/mol) or *transoid-TS4* (3.4 kcal/mol), then release **3a** after keto-enol tautomerism.” “Similarly, the introduction of **1a** to *cisoid-E_{CO}* initiates the proton transfer via transition state *cisoid-TS5* (6.5 kcal/mol), leading to **3a** after keto-enol tautomerism”

Fig. 9. Computed potential energy surface for the Mn(I)-catalyzed carbon-skeleton rearrangement via C-C bond activation at the SMD(DCE)/M06L/Def2tzvp//M06L/def2svp level of theory.

Question 4: Also, the authors did not investigate how the active species is formed from $\text{MnBr}(\text{CO})_5$ and **1a**. In this step, the C-H sigma bond activation is involved. It is likely that the reaction does not occur with small activation energy. It is important to show how much easily the active species is formed.

Respond to this question: We calculated the formation of the active species from the $\text{Mn}(\text{CO})_5\text{Br}$ and **1a** (see Fig. S8). The first step is the ligand-exchange of $\text{Mn}(\text{CO})_5\text{Br}$ generating the six-coordination intermediates *cisoid-A_{pre}* (-13.4 kcal/mol) or *transoid-A_{pre}* (-12.1 kcal/mol), in which the pyridine coordinate with Mn center. Then, with the assistance of the bromide, the alcohol is deprotonated via transition states *cisoid-TS6* or *transoid-TS6*, respectively. The Gibbs free energy of activation of *cisoid-TS6* is about 5.8 kcal/mol, which is lower than that of *transoid-TS6* by about 2.5 kcal/mol. As the reviewer commented, the formation of the active species has to overcome relatively high Gibbs free energy of activation. We have added this discussion in SI.

Fig. S8. Computed potential energy surface for the reaction between $\text{Mn}(\text{CO})_5\text{Br}$ and 1a at the SMD(DCE)/M06L/Def2tvzp//M06L/def2svp level of theory.

Question 5: The analysis shown in Fig. 10 is not reasonable. This is the insertion reaction of the C=C or C=O double bond into the Mn-alkyl bond. In Fig. 10, the authors explained that the charge-transfer (CT) from Mn d-pai orbital to the C=C (or C=O) pai* orbital is important for this transition state. However, this CT does not lead to the formation of the C-C bond. In the transition state of the insertion reaction, the sp² orbital of the vinyl group, which interacts with the Mn empty d-sigma orbital in the intermediate B, starts to interact with the pai* orbital of the C=C (or C=O) double bond, which participates in the pai-back donation interaction with the Mn atom in the intermediate B. We must consider the possibility that the activation energy would become larger if the pai-back donation is strong in the intermediate B. In other words, the strong pai-back donation is not always favorable for lowering the activation energy. The schematical figure of Fig. 10 is not appropriate for discussion and the discussion must be corrected.

Respond to this question: Thanks for your good comments. The C=O and C=C insertion steps were not the charge-transfer (CT) from Mn d orbital to the C=C (or C=O) π^* orbital. We reanalyzed the insertion of the C=C or C=O double bond into the Mn-alkyl bond shown in **Fig. 10**.

Actually, the reverse β -aryl elimination reaction (C=O insertion) is driven by 1) the interaction between the C-Mn p-d orbital and the C=O π^* antibonding orbital, and 2) the interaction between the C=O π orbital and the Mn d_{z^2} anti-bonding orbital. In comparison, the C=C insertion is contributed by the donation of the C-M p-d orbital to the C=C π^* orbital and the donation of the C=C π orbital to the antibonding d_{z^2} orbital. In both cases, the energy gaps between the C-M p-d orbital and the π^* orbital (3.10 eV for the C=O insertion, 3.05 eV for the C=C insertion) are smaller than those between the π orbital and the antibonding d_{z^2} orbital (4.46 eV for the C=O insertion, 4.54 eV for the C=C insertion), suggesting the p-d to π^* interaction attributes mainly for the insertion. Notably, the p-d to π^* interaction is more significant in the C=C insertion ($\Delta E_{\text{gap}} = 3.05$ eV) than in the C=O insertion ($\Delta E_{\text{gap}} = 3.10$ eV), resulting in a better interaction energy ($\Delta E_{\text{int}} = -14.3$ kcal/mol) in the C=C insertion transition state **transoid-TS2**, which is probably the driving

force of the preference of C=C insertion over C=O insertion for the rearrangement process.

Fig. 10. The interaction between metal-carbon π bond and LUMO on the acrylaldehyde, C=O and C=C π antibonding orbital of acrylaldehyde and the empty d orbital of Mn atom, and deformation/interaction analysis of the transition states for the β -aryl elimination pathway and migratory insertion pathway. Energies are in kcal/mol, isovalue of Kohn-Sham orbitals 0.05.

Question 6: Page 15 the last line to page 16 the first line: The orbital energy gaps discussed here show small difference (0.05 eV). In Figure 10, the Delta E(gap) is presented below the Kohn-Sham orbital at the transition state. In my understanding, this is the energy gap between the Mn occupied d orbital of the separated Mn moiety and the π^* orbital of the C=O or C=C bond of the separated substrate. The definition must be presented.

Respond to this question: As shown in Fig.10, it is the energy gap between the metal-carbon d-p orbital of the separated Mn moiety and the LUMO of the acrylaldehyde as separated substrate. And this definition has been presented in the revised manuscript.

Question 7: The other factor for determining the regioselectivity is the orbital shape of the π^* orbital. The authors must compare the size of terminal C p orbital and the carbonyl C p orbital in the LUMO of the substrate. The difference in size leads to the difference in regioselectivity.

Respond to this question: Thank you for the good suggestion. We further analyzed the orbital composition in LUMO of the substrate shown in Fig. 10. The calculated results indicate that the terminal C⁴ p orbital (32.9%) in C=C insertion step is greater than the carbonyl C² p orbital (27.5%) in C=O insertion step, resulting in a large driving force for the C=C insertion, guaranteeing the subsequent group-inversion rearrangement. As pointed out nicely by the referee, the difference in the orbital composition leads to the difference in regioselectivity.

We have also added this discussion to the revised manuscript "Further the orbital composition analysis in the LUMO of the substrate indicates that the terminal C⁴ p orbital (32.9%) in C=C insertion step is greater than the carbonyl C² p orbital (27.5%) in C=O insertion step, resulting in a large driving force for the C=C insertion, guaranteeing the subsequent group-inversion rearrangement."

Question 8: Minor comments: Several words are not correctly used. The authors must ask the physical and/or theoretical chemist to check the sentences. "activation free energy"; There are two kinds of free energy. Here, either "the Gibbs energy of activation" or "the Gibbs free energy of activation" is correct. Or, the Gibbs activation energy is useful, though this is not a correct name. Back-bonding: What is the chemical bond toward back-side? The "back-donation" must be used.

Respond to this question: We have modified "activation free energy" to "the Gibbs free energy of activation" and corrected the "back-donation".

Yours Sincerely

Prof. Wei Zeng
School of Chemistry and Chemical Engineering
South China University of Technology
Guangzhou 510641, P. R. China

REVIEWERS' COMMENTS

Reviewer #1 (Remarks to the Author):

The resubmitted manuscript has been appropriately revised with additional data to clearly explain the obtained results. However, the newly used "U-Turn" rearrangement is completely unclear; "U-turn" is not defined. The 1,4-adduct is not "the opposite direction" of the 1,2-adduct. If there is no clear explanation, this term should not be used to avoid confusing the classification of rearrangement reactions.

Reviewer #2 (Remarks to the Author):

The authors have made substantial supplementary studies by both experimental and computational approaches to answer the concerns from the reviewers. Most of them have been well addressed. The revised manuscript was considerably improved. Reviewer 2 recommend it for publication in Nature communications after the following new comments being noted.

The reactions disclosed in this study involves several organometallic transformative steps that leads to skeleton arrangement of the substrates, which is not unique in organic chemistry. The reviewer does not appreciate the way to invent new concepts for arrangement reactions unless it is fundamentally novel. I suggest the authors to modify the title slightly by using "U-turn-like Rearrangement" or "skeleton Rearrangement".

The reviewer agrees with the response regarding the difference of this study to previously reported reactions (J. Am. Chem. Soc. 2007, 129, 46, 14158; Angew. Chem., Int. Ed. 2021, 60, 20190). However, as they achieved formally very similar arrangements, they should be cited with appropriate discussion in the main text.

Reviewer #4 (Remarks to the Author):

As a computational chemist, I have been asked by the associate editor of Nature Communications to evaluate the revised version – especially the modelling contribution – of the manuscript "NCOMMS-

22-27391A-Z // Mn(I)-Catalyzed In-Situ Group-Inversion Rearrangement" submitted for publication by Profs Ke and Zeng.

In the point-by-point answer letter, the authors provided constructive and convincing arguments to most of the question raised at the first round of evaluation.

Regarding the influence of the density functional on computed energy profiles, the combination of highly parametrized Minnesota functionals with empirical dispersion is odd. The comparison with B3LYP(D3) or PBE0(D3) is more informative. The use of the Becke-Johnson damping function brings more precise dispersion correction. Additionally, the use of an ultrafine integration grid while using Minnesota functionals is also relevant to avoid artifact. However, doing these refinements will most likely does not change the mechanistic picture afforded by the authors as trends are mostly maintained and I am not asking for more.

Regarding the Gibbs energy profile provided in Figure 9. I find it overloaded and difficult to read. I suggest to the authors to simplify it and to give the full version in SI, or to find a way to enhance its readability. In the caption, please indicate that energies are indeed ΔG values.

The caption of Figure 10 is more a comment on the figure than a title. Please correct it. Distance unit is missing. Additionally, the readability and the self-understanding of this figure must be raised.

Regarding the SI, authors did insert Cartesian coordinates of optimized structures, but associated energies are missing. please add them.

Response to Referees

Journal: *Nat. Commun.*

Manuscript ID: NCOMMS-22-27391A-Z

Title: Mn(I)-Catalyzed U-Turn-Like-Rearrangement of β , γ -Unsaturated Alcohols

Author(s): Can Yang, Xiaoyu Zhou, Lixing Shen, Zhuofeng Ke*, Huanfeng Jiang, and Wei Zeng*

Address: ^a Key Laboratory of Functional Molecular Engineering of Guangdong Province, School of Chemistry and Chemical Engineering, South China University of Technology, Guangzhou 510641, China; ^b School of Materials Science and Engineering, PFCM Lab, Sun Yat-sen University, Guangzhou 510275, China

Dear Referees,

The above-mentioned manuscript was ever submitted to *Nat. Commun.(NC)* for peer review. On Oct. 15th, 2022, and we got the review results from you on Jan. 2nd, 2023. Now we have already revised our manuscript according to the referees' suggestions, Some revisions are shown as follows, please check them.

A. Respond to the 1st reviewer's comments

Question 1: However, the newly used "U-Turn" rearrangement is completely unclear; "U-turn" is not defined. The 1,4-adduct is not "the opposite direction" of the 1,2-adduct. If there is no clear explanation, this term should not be used to avoid confusing the classification of rearrangement reactions.

Respond to this question: We already gave a "U-Turn" definition in our revised manuscript, this concept refers to a U-shaped turn made by a molecular skeleton so as to head in the opposite direction from its original course. Moreover, considering the 2nd reviewer's suggestion, we modified the manuscript title slightly by using "U-turn-like" instead of "U-Turn", please check it, thanks.

B. Respond to the 2nd reviewer's comments

Question 1: The reviewer does not appreciate the way to invent new concepts for arrangement reactions unless it is fundamentally novel. I suggest the authors to modify the title slightly by using "U-turn-like Rearrangement" or "skeleton Rearrangement".

Respond to this question: Thanks for the 2nd reviewer's suggestion, we already modified the manuscript title slightly by using "U-turn-like" instead of "U-Turn". Moreover, we utilized the symbol "UT-R" to represent the abbreviation of "U-turn-like Rearrangement" in our whole manuscript, please check it, thanks.

Question 2: The reviewer agrees with the response regarding the difference of this study to previously reported reactions (*J. Am. Chem. Soc.* 2007, 129, 46, 14158; *Angew. Chem., Int. Ed.* 2021, 60, 20190). However, as they achieved formally very similar arrangements, they should be cited with appropriate discussion in the main text.

Respond to this question: We already cited these two papers (see ref. 51: *Angew. Chem., Int. Ed.* 2021, 60, 20190; see ref. 53: *J. Am. Chem. Soc.* 2007, 129, 14158;), and briefly discussed the corresponding work in our revised manuscript as follows:

“..., Comparison with the photocatalyzed 1,3-alkyl shift of tertiary allylic alcohols reported by Knowles⁵¹, our catalysis system could be applied to tertiary allylic alcohols **4** which would undergo chemoselective 1,3-aryl transposition (**Fig. 3**). Among them, α -(2-indolyl)- α -alkyl-allylic alcohols (**4a-4c**) and...”

“..., Encouraged by Rh(I)-catalyzed 1,3-alkynyl shift of alkynyl alkenyl carbinols in which Csp³-Csp bond cleavage occurred⁵³, we further evaluated the rearrangement reactivity of different α -(2-indolyl)propargyl alcohols (**Fig. 4**). Interestingly,...”

C. Respond to the 4th reviewer’s comments

Question 1: Regarding the influence of the density functional on computed energy profiles, the combination of highly parametrized Minnesota functionals with empirical dispersion is odd. The comparison with B3LYP(D3) or PBE0(D3) is more informative. The use of the Becke–Johnson damping function brings more precise dispersion correction. Additionally, the use of an ultrafine integration grid while using Minnesota functionals is also relevant to avoid artifact. However, doing these refinements will most likely does not change the mechanistic picture afforded by the authors as trends are mostly maintained and I am not asking for more.

Respond to this question: Thank you again for the reviewer’s very constructive comments on the first-round revision. We further evaluated other density functionals like B3LYP and PBE0 with the combination of Becke-Johnson Damping (D3-BJ) functions. We also used the ultrafine integration grid for Minnesota functionals to avoid artifacts, as well as other functionals.

All the structures optimization and frequency calculations using the ultrafine integration grid were done with the M06-L, M06-L(D3), M06, M06(D3), B3LYP(D3BJ), PBE0(D3BJ), and MN15 functional. All the results suggest that the Gibbs activation energies of the β -aryl elimination step (the *cisoid-TS1* and *transoid-TS1*) are also higher than those of the C=C insertion (the *cisoid-TS2* and *transoid-TS2*) and the 1,4-addition (*cisoid-TS3*) steps. For the β -aryl elimination step, the Gibbs activation energies with B3LYP(D3BJ), PBE0(D3BJ), and MN15 functional are slightly higher than that with M06-L, M06(D3) functional. while the Gibbs activation energies of the C=C insertion via the *cisoid-TS2* and *transoid-TS2* with B3LYP(D3BJ) functional are slightly higher than that with M06-L, M06(D3), PBE0(D3BJ), and MN15 functional. The Gibbs activation energy of the *cisoid-TS3* with B3LYP(D3BJ) functional is also lower than that of PBE0(D3BJ) and MN15 functional. The results suggest that Minnesota functional (M06-L, M06, and MN15) can well present the dispersion effect. The calculated results with M06-L, M06, and MN15 functional are consistent with the experimental results.

Overall, the DFT computational results with B3LYP(D3BJ) and PBE0(D3BJ) functional and Minnesota functional with ultrafine integration grid do not change the mechanism and the discussion.

We have put these results in **Supplementary Table 8** in the Supporting information.

	M06-L	M06-L(D3)	B3LYP(D3BJ)	PBE0(D3BJ)	MN15	M06(D3)	M06
cisoid-A	0.0	0.0	0.0	0.0	0.0	0.0	0.0
transoid-A	2.5	2.5	0.5	0.4	1.6	0.4	1.2
cisoid-TS1	16.2	16.6	18.9	18.8	18.2	16.3	15.4
transoid-TS1	18.5	18.8	20.3	19.9	19.4	17.2	16.7

cisoid-B	1.4	1.1	0.5	3.3	4.5	1.4	0.7
transoid-B	-1.2	-0.4	-1.9	1.1	3.8	0.0	-2.0
cisoid-C	4.8	5.2	6.9	5.1	4.6	2.1	1.3
transoid-C	2.7	3.2	4.9	3.5	2.7	0.7	0.2
cisoid-TS2	11.4	10.6	13.9	13.1	12.3	8.0	7.5
transoid-TS2	10.3	11.9	12.3	12.1	10.9	7.0	6.3
cisoid-TS3	18.7	18.8	15.5	25.4	23.0	19.2	18.8

Question 2: Regarding the Gibbs energy profile provided in Figure 9. I find it overloaded and difficult to read. I suggest to the authors to simplify it and to give the full version in SI, or to find a way to enhance its readability. In the caption, please indicate that energies are indeed DeltaG values.

Respond to this question: We have simplified Fig. 9 focusing on the major reaction pathways (Note: the previous Fig. 9 was already changed Fig. 8 in our revised manuscript, please check it), and the full version is now present in Supplementary Fig. 8. The DeltaG has been marked in Fig. 8, Supplementary Fig. 8, and Supplementary Fig. 9, as well as in the captions.

Fig. 8. Computed potential energy surfaces for the major reaction pathways of the Mn(I)-catalyzed carbon-skeleton rearrangement *via* C-C bond activation at the SMD(DCE)/M06L/Def2tzvp//M06L/def2svp level of theory. Free energies are in kcal/mol.

Supplementary Fig. 8. Computed potential energy surface for the Mn(I)-catalyzed carbon-skeleton rearrangement *via* C-C bond activation at the SMD(DCE)/M06L/Def2tzvp//M06L/def2svp level of theory. Free energies are in kcal/mol.

Question 3: The caption of Figure 10 is more a comment on the figure than a title. Please correct it. Distance unit is missing. Additionally, the readability and the self-understanding of this figure must be raised.

Respond to this question: We have corrected the title and updated Fig. 10 as follows (Note: Fig. 10 was already changed to Fig. 9), please check it, thanks.

Fig. 9. The localized Kohn-Sham orbitals and distortion/interaction analysis. **a** C-Mn p-d orbital and C=O π^* antibonding orbital interaction, C=O π orbital and Mn d_{z^2} orbital interaction. **b** Deformation/interaction analysis of the transition state for the β -aryl elimination pathway. **c** C-M p-d orbital and C=C π^* orbital interaction, C=C π orbital and Mn d_{z^2} orbital interaction

d Deformation/interaction analysis of the transition state for the migratory insertion pathway. Free energies are in kcal/mol, isovalue of Kohn-Sham orbitals is 0.05.

Question 4: Regarding the SI, authors did insert Cartesian coordinates of optimized structures, but associated energies are missing, please add them.

Respond to this question: The absolute energies of all optimized structures are present in Supplementary Table 9 in SI.

Supplementary Table 9 Absolute energies of all optimized structures

Species	E_c	H_c	G_c	$G_{QT=383.585}$	$E/(\text{Hartree})$	$H/(\text{Hartree})$	$G/(\text{Hartree})$	Solvation	$G_{sol}/(\text{Hartree})$
								Energy/(Hartree)	
Mn(CO) ₅ Br	0.043528	0.058250	0.002114	-0.007300	-4290.830790	-4290.816068	-4290.872204	-4292.005367	-4292.012668
CO	0.005078	0.008383	-0.014052	-0.017815	-113.203742	-113.200437	-113.222872	-113.338245	-113.356061
HBr	0.006080	0.009385	-0.013143	-0.016920	-2574.346434	-2574.343129	-2574.365656	-2574.668463	-2574.685383
1a	0.292213	0.310442	0.247014	0.236377	-841.146017	-841.127788	-841.191217	-842.351544	-842.115167
3a	0.292301	0.310416	0.246349	0.235606	-841.178914	-841.160799	-841.224866	-842.384847	-842.149241
cisoid-A	0.315735	0.343906	0.257579	0.243101	-2444.385984	-2444.357813	-2444.444140	-2446.291606	-2446.048505
transoid-A	0.316206	0.344215	0.258646	0.244296	-2444.381845	-2444.353836	-2444.439405	-2446.288885	-2446.044589
cisoid-A1	0.308001	0.333271	0.254297	0.241053	-2331.151046	-2331.125776	-2331.204750	-2332.918748	-2332.677695
transoid-A1	0.307914	0.333398	0.253404	0.239989	-2331.146435	-2331.120952	-2331.200946	-2332.915539	-2332.675550
cisoid-TS1	0.305984	0.331287	0.251835	0.238511	-2331.140625	-2331.115323	-2331.194775	-2332.905070	-2332.666560
transoid-TS1	0.306135	0.331263	0.252771	0.239608	-2331.137416	-2331.112288	-2331.190780	-2332.902583	-2332.662975
cisoid-B	0.305878	0.332544	0.248502	0.234408	-2331.158405	-2331.131740	-2331.215782	-2332.924683	-2332.690275
transoid-B	0.305981	0.332568	0.248758	0.234703	-2331.161113	-2331.134525	-2331.218336	-2332.929109	-2332.694406
cisoid-C	0.306832	0.332647	0.252458	0.239010	-2331.155721	-2331.129905	-2331.210095	-2332.923822	-2332.684813
transoid-C	0.306942	0.332716	0.251989	0.238452	-2331.156934	-2331.131161	-2331.211887	-2332.926525	-2332.688073
cisoid-TS2	0.306945	0.331866	0.254124	0.241086	-2331.148320	-2331.123399	-2331.201142	-2332.915325	-2332.674239
transoid-TS2	0.306890	0.331803	0.253791	0.240709	-2331.147688	-2331.122775	-2331.200787	-2332.916802	-2332.676093
cisoid-TS3	0.306280	0.331271	0.253421	0.240366	-2331.141054	-2331.116063	-2331.193912	-2332.902979	-2332.662613
cisoid-D	0.308885	0.333836	0.256265	0.243257	-2331.174379	-2331.149428	-2331.226998	-2332.941030	-2332.697774
cisoid-D _{CO}	0.317912	0.345275	0.262128	0.248185	-2444.410088	-2444.382725	-2444.465871	-2446.316484	-2446.068299
transoid-D	0.308211	0.333462	0.254535	0.241299	-2331.167321	-2331.142070	-2331.220997	-2332.938458	-2332.697159
transoid-D _{CO}	0.317509	0.345033	0.261225	0.247171	-2444.406354	-2444.378831	-2444.462638	-2446.316614	-2446.069443
cisoid-E	0.308292	0.333594	0.254877	0.241676	-2331.169358	-2331.144055	-2331.222772	-2332.933135	-2332.691459
cisoid-E _{CO}	0.316443	0.344308	0.259317	0.245065	-2444.402086	-2444.374222	-2444.459212	-2446.307026	-2446.061961
cisoid-TS4	0.610282	0.655504	0.532012	0.511302	-3285.568685	-3285.523462	-3285.646955	-3288.668236	-3288.156934
transoid-TS4	0.608329	0.653798	0.528859	0.507906	-3285.569671	-3285.524202	-3285.649141	-3288.666203	-3288.158297
cisoid-TS5	0.606568	0.652351	0.526853	0.505807	-3285.562126	-3285.516343	-3285.641841	-3288.658472	-3288.152665

cisoid-TS6	0.328427	0.358498	0.267291	0.251995	-5018.739804	-5018.709733	-5018.800940	-5020.976572	-5020.724577
cisoid-A _{ppc}	0.329348	0.360113	0.267567	0.252046	-5018.772845	-5018.742080	-5018.834627	-5021.007271	-5020.755225
transoid-TS6	0.328874	0.358745	0.268699	0.253598	-5018.737661	-5018.707791	-5018.797836	-5020.974321	-5020.720723
transoid-A _{ppc}	0.329534	0.360269	0.267977	0.252500	-5018.770656	-5018.739922	-5018.832213	-5021.005711	-5020.753211

Now, we already revised our manuscript according to all the reviewers' suggestions, and then re-submit it to *Nat. Commun.* If any questions, please let me know, thanks very much.

Yours Sincerely

Prof. Wei Zeng
School of Chemistry and Chemical Engineering
South China University of Technology
Guangzhou 510641, P. R. China